# Pollutant emission reductions deliver decreased $PM_{2.5}$-caused mortality across China during 2015-2017

**Ben Silver[1], Luke Conibear[1], Carly L. Reddington[1], Christoph Knote[2], Steve R. Arnold[1] and Dominick V. Spracklen[1]**

[1] School of Earth and Environment, University of Leeds, Leeds, LS2 9JT, UK
[2] Meteorological Institute, Ludwig-Maximilians-University Munich, Theresienstr. 37, Munich, 80333, Germany

*Correspondence to*: Ben Silver (eebjs@leeds.ac.uk)

**Abstract.** Air pollution is a serious environmental issue and leading contributor to the disease burden in China. Rapid reductions in fine particulate matter ($PM_{2.5}$) concentrations and increased ozone concentrations have occurred across China, during 2015 to 2017. We used measurements of particulate matter with a diameter $< 2.5$ µm ($PM_{2.5}$) and Ozone ($O_3$) from >1000 stations across China along with Weather Research and Forecasting model coupled with Chemistry (WRF-Chem) regional air quality simulations, to explore the drivers and impacts of observed trends. The measured nationwide median $PM_{2.5}$ trend of -3.4 µg m$^{-3}$ year$^{-1}$, was well simulated by the model (-3.5 µg m$^{-3}$ year$^{-1}$). With anthropogenic emissions fixed at 2015-levels, the simulated trend was much weaker (-0.6 µg m$^{-3}$ year$^{-1}$), demonstrating interannual variability in meteorology played a minor role in the observed $PM_{2.5}$ trend. The model simulated increased ozone concentrations in line with the measurements, but underestimated the magnitude of the observed absolute trend by a factor of 2. We combined simulated trends in $PM_{2.5}$ concentrations with an exposure-response function to estimate that reductions in $PM_{2.5}$ concentrations over this period have reduced $PM_{2.5}$-attribrutable premature morality across China by 150 000 deaths year$^{-1}$.

## 1 Introduction

Concentrations of particulate matter and ozone across China largely exceed international air quality standards (Reddington et al., 2019; Silver et al., 2018). This poor air quality is estimated to hasten the deaths of 870 000 - 2 470 000 people across China each year (Apte et al., 2015; Burnett et al., 2018; Cohen et al., 2017; Gu and Yim, 2016; Lelieveld et al., 2015). The Chinese government's efforts to improve air quality began in the 1990s, but emissions of pollutants continued to increase into the 21$^{st}$ century and air pollution worsened (Krotkov et al., 2016; Streets et al., 2008; Zhang et al., 2012). In 2013, China experienced episodes of severe particulate matter pollution (Zhang et al., 2016). In response, the Chinese government announced the Action Plan on the Prevention and Control of Air Pollution which focused on the reduction of fine particulate matter ($PM_{2.5}$) through stringent emission controls during 2012-2017 (Zheng et al., 2017).

## 1.1 Previous studies of trends in China's air quality

Satellite remote sensing studies have been used to show large changes in air pollution across China in recent decades, with positive trends in Nitrogen Dioxide ($NO_2$) (Van der A et al., 2006), Sulfur Dioxide ($SO_2$) (Zhang et al., 2017) and $PM_{2.5}$ (Ma et al., 2016) during the 1990s and early 2000s. Trends in aerosol optical depth have been used to estimate changes in $PM_{2.5}$, which peaked around 2011 (Ma et al., 2016). $NO_2$ across China peaked around 2011 (De Foy et al., 2016; Irie et al., 2016), although concentrations in the Pearl River Delta (PRD) peaked earlier and western regions may have peaked later (Cui et al., 2016). Several remote sensing studies show that $SO_2$ concentrations in China peaked around 2006 (Van Der A et al., 2017; Krotkov et al., 2016; Zhang et al., 2017), matching the period of maximum emissions (Duan et al., 2016; Li et al., 2017a; Zheng et al., 2018). Analysis of measurements from the Acid Deposition Monitoring Network in East Asia (EANET) shows a negative pH trend (i.e., becoming more acidic) from 1999 until a reversal occurs in 2006, matching peak $SO_2$ emissions and concentrations (Duan et al., 2016). Measurements of $O_3$ concentrations at background monitoring sites indicate positive trends in western China during 1994-2013 (Xu et al., 2016), and Taiwan during 1994-2003 (Chang and Lee, 2007), suggesting that $O_3$ has been increasing across China during the past two decades. More recently, measurements at urban sites, also show positive $O_3$ trends during 2005-2011 (Zhang et al., 2014).

The establishment of China's air pollution monitoring network, operated by the China National Environmental Monitoring Centre (CNEMC) (Wang et al., 2015), which includes measurements from over 1600 locations, has enabled more detailed analysis of recent air pollution changes (Silver et al., 2018; Zhai et al., 2019). Between 2015 and 2017, $PM_{2.5}$ concentrations across China decreased by 28% (Silver et al., 2018). Zhai et al., (2019) reported a 30-40% decrease in $PM_{2.5}$ concentrations during 2013-2017. In contrast $O_3$ concentrations have increased, with median concentration of $O_3$ across 74 key cities increasing from 141 $\mu g\ m^{-3}$ in 2013 to 164 $\mu g\ m^{-3}$ in 2017 (Huang et al., 2018). Silver et al. (2018) found that $O_3$ maximum 8 h mean concentrations ($O_3MDA8$) increased by 4.6 % $year^{-1}$ over 2015-2017. Lu et al., (2020) reported positive trends in April-September $O_3MDA8$ at 90% of sites during 2013 to 2019. Positive regional $O_3$ trends remain even after meteorological variability has been removed (Li et al., 2019b). Trends in $NO_2$ are more variable, with a negative trend reported in eastern China and positive trends in western areas (Li and Bai, 2019). Silver et al., (2018) found that $NO_2$ had negative trends in Hong Kong and North China Plain regions, but positive trends in the Yangtze River Delta (YRD), Sichuan Basin (SCB) and PRD, and no overall trend at the national scale.

## 1.2 Identifying drivers of recent trends

Changes in the concentrations of air pollutants may be caused by changing emissions or by interannual variability of meteorology. Stringent emission controls have started to reduce emissions of various pollutants across China. Between 2013 and 2017, emissions of $PM_{2.5}$, $SO_2$ and $NO_x$ ($NO_2$ + Nitrogen Oxide) declined whereas emissions of Ammonia ($NH_3$) and Non-Methane Volatile Organic Compounds (NMVOCs) remained fairly constant (Zheng et al., 2018). B. Zheng et al. (2018) also demonstrate that emission reductions were primarily driven by pollution controls, rather than decreasing activity rates.

Meteorological variability alters atmospheric mixing, deposition and transport, all of which can influence the concentration of pollutants. Separating the influence of meteorology and emissions on air pollutant concentrations is difficult, due to the interlinked nature of the chemistry-climate system (Jacob and Winner, 2009). However, to assess the efficacy of China's emissions reductions, it is necessary to separate these two factors.

There are two commonly used approaches to separate the influences of meteorology and emissions on variability in atmospheric pollutant abundances. The first approach uses statistical models, such as multi-linear regression, to control for the influence of meteorology and allowing the proportion of air pollutant concentration variability that can be explained by meteorological variables to be calculated (Tai et al., 2010). The second approach is to use an atmospheric chemistry transport model to simulate pollutant concentrations (Ansari et al., 2019; Xing et al., 2011).

There are a limited number of modelling studies that attempt to separate the influence of meteorology and emissions changes on recent air quality trends in China. Chen et al. (2019) use WRF-Chem with 2010 emissions to examine the drivers of trends in wintertime PM. Ding et al. (2019) use WRF-CMAQ to evaluate importance of emissions, meteorology and demographic changes on $PM_{2.5}$ related mortality during 2013-2017. Our paper adds to these previous studies by evaluating the ability of a online-coupled model (WRF-Chem) to capture trends in $NO_2$, $O_3$ and $SO_2$ as well as PM, using the most recent emissions and evaluated against a comprehensive measurement dataset.

Through a comparison of multiple simulations, where either annual variability in emissions or meteorology are held constant, the relative influence of the two factors can be estimated. Here we analyse measurements and a regional air quality model to explore the role of changing anthropogenic emissions on air pollutant concentrations and human health across China during 2015 to 2017.

## 2 Materials and Methods

### 2.1 Measurement dataset

We used hourly measurements from the CNEMC monitoring network (Wang et al., 2015) of $PM_{2.5}$, $O_3$, $NO_2$, and $SO_2$ for the period 2015-2017, which includes data from over 1600 monitoring stations across mainland China and is available to download from http://beijingair.sinaapp.com/. This was combined with data from the Hong Kong Environmental Protection Department (https://cd.epic.epd.gov.hk/EPICDI/air/station/) and Taiwan's Environmental Protection Administration (https://taqm.epa.gov.tw/taqm/en/YearlyDataDownload.aspx). We conducted quality control on the measured data following the methods outlined in Silver et al. (2018), which include excluding data with a high proportion of repeated measurements and periods of low variability. The cleaned dataset included measurements from 1155 sites.

### 2.2 WRF-Chem model setup

We used the Weather Research and Forecasting model with Chemistry (WRF-Chem) version 3.7.1 (Grell et al., 2005) to simulate trace gas and particulate pollution over China for 2015 to 2017. The model domain uses a Lambert Conformal grid

(11-48 °N, 93-128 °E) centred on eastern China with a horizontal resolution of 30 km. The model has 33 vertical layers, with the lowest layer ~29 m above the surface, and the highest at 50 hPa (~19.6 km).

European Centre for Medium Range Weather Forecasts (ECMWF) ERA-Interim fields were used to provide meteorological boundary and initial conditions, as well to nudge the model temperature, winds and humidity above the boundary layer every 6 hours. Restricting nudging to above the boundary layer, allowed a more realistic representation of vertical mixing (Otte et al., 2012). Chemical boundary and initial conditions were provided by global fields from the Model for Ozone and Related Chemical Tracers version 4 (MOZART-4) chemical transport model (Emmons et al., 2010).

Anthropogenic emissions were from the Multi-resolution Emission Inventory for China (MEIC; www.meicmodel.org). MEIC estimates emissions using a database of activity rates across residential, industrial, electricity generation, transportation and agricultural emission sectors combined with China-specific emission factors (Hong et al., 2017). We used the 2015 MEIC dataset, then used sector-specific and species-specific scaling for 2016 and 2017 based on the emission totals estimated in B. Zheng et al. (2018). Table 1 shows emission totals for 2015, 2016 and 2017. Over the 2015 to 2017 period, Chinese emissions decreased by 38% for $SO_2$, 16% for $PM_{2.5}$ and 8% for NOx. For regions outside the MEIC dataset, we used anthropogenic emissions from the EDGAR-HTAPv2.2 emission inventory for 2010.

Biogenic emissions were generated online by the Model of Emissions of Gases and Aerosol from Nature (MEGAN) (Guenther et al., 2000). Biomass burning emissions were provided by the Fire Inventory from NCAR (FINN) version 1.5 (Wiedinmyer et al., 2011), which uses satellite fire observations of fires and land cover to estimate daily 1 $km^2$ emissions. Dust emissions were generated online the Georgia Institute of Technology-Goddard Global Ozone Chemistry Aerosol Radiation and Transport (GOCART) model with Air Force Weather Agency (AFWA) modifications (LeGrand et al., 2019).

Gas-phase chemistry is simulated using the MOZART-4 scheme and aerosol is treated by the Model for Simulating Aerosol Interactions and Chemistry (MOSAIC; Zaveri *et al.*, 2008) scheme, including grid-scale aqueous chemistry and an extended treatment of organic aerosol (Hodzic and Jimenez, 2011; Hodzic and Knote, 2014). Four discrete size bins were used within MOSAIC (0.039–0.156 μm, 0.156–0.625 μm, 0.625–2.5 μm, 2.5–10 μm) to represent the aerosol size distribution.

**2.3 Model and measurement trend estimation**

For comparison with the measurements, we sampled the model at the station locations using linear interpolation. Over 2015-2017, the model well simulated $PM_{2.5}$ (normalised mean bias (NMB) = 0.45), $O_3$ (NMB=-0.13) and $SO_2$ (NMB=0.07), while overestimating $NO_2$ concentrations by a factor of around 2 (NMB=1.17). Model biases were similar to previous model studies in China (Supplementary Table 1). We also evaluated the model against speciated aerosol measurements from the Surface PARTiculate mAtter Network (SPARTAN) (Snider et al., 2015, 2016) site in Beijing (https://www.spartan-network.org/beijing-china, last accessed: 2nd July 2020) (Fig S4), as well as Zhou et al. (2019) (Figure S5) and from across China (Li et al., 2017b) (Fig S6). Measurements reported by Li et al. (2017b) were made from various years spanning 2006 to 2013 and do not match the years simulated by the model. Comparison against these data show that the model underestimates the sulfate fraction in $PM_{2.5}$, while overestimating the nitrate fraction. Underestimation of sulfate in comparison to Li et al.,

(2017b) will partly be caused by the large decline in $SO_2$ emissions that has occurred in the last decade (Zheng et al., 2018).

Underestimate of sulfate, particularly in winter, and overestimation of nitrate are consistent with previous modelling studies (Shao et al., 2019)   including those using WRF-chem (Zhou et al., 2019). Newly proposed mechanisms to explain the rapid sulfate formation in China's winter haze (Gen et al., 2019; Shao et al., 2019; Xue et al., 2014; Zhang et al., 2019) need to be included and evaluated in models.

To separate the influence of changing anthropogenic emissions from interannual variability in meteorology, we conducted two

3-year simulations, both for 2015-2017. The first simulation (Control) included interannual variability in both anthropogenic emissions and meteorology. The second simulation (Fixed emissions) included interannual variability in meteorology, but with anthropogenic emissions fixed at 2015 levels. Both simulations include interannual variability in biogenic and biomass burning emissions, allowing us to isolate the impacts of changing anthropogenic emissions.

Trends in the model data were calculated using the same method as the measurement data (Silver et al., 2018). The hourly data

are averaged to monthly means, which are then deseasonalised using locally weighted scatterplot smoothing. The magnitude and direction of linear trends were calculated using the Theil-Sen estimator, a non-parametric method that is resistant to outliers (Carslaw, 2015). The Mann-Kendall test was used to assess the significance of trends, using a threshold of $p < 0.05$. This stage of the analysis was performed using the R package '*openair*' (Carslaw and Ropkins, 2012).

## 2.4 Health impact estimation

Health impacts are estimated for ambient $PM_{2.5}$ using the Global Exposure Mortality Model (GEMM) (Burnett et al., 2018), which uses cohort studies to estimate health risks integrated over a range of $PM_{2.5}$ concentrations. GEMM applies a supralinear association between exposure and risk at lower concentrations and then a near-linear association at higher concentrations. We used the GEMM for non–accidental mortality (non–communicable disease, NCD, plus lower respiratory infections, LRI), using parameters including the China cohort (GBD 2017 Risk Factor Collaborators, 2018). For ambient $O_3$, we used the

methodology of the Global Burden of Disease (GBD) study for 2017 (GBD 2017 Risk Factor Collaborators et al., 2018) to estimate the mortality caused by chronic obstructive pulmonary disease, which is based on exposure and risk information from five epidemiological cohorts. It estimates a near-linear relationship between exposure and risk at lower concentrations of $O_3$, and a sub-linear association at higher concentrations. The United Nations adjusted population count dataset for 2015 at $0.05°$ $\times 0.05°$ resolution was obtained from the Gridded Population of the World, Version 4, along with population age distribution

from GBD2017. Health impacts depend on population count, population age, and baseline mortality rates which have changed over the period studied (Butt et al., 2017). To isolate the impacts of changing air pollution, other variables were kept constant for 2015-2017.

# 3 Measured and modelled trends comparison

## 3.1 Varying emissions scenario

Figure 1 and 2 compare measured and simulated air quality trends over China during 2015 to 2017. The measurements show widespread decline in $PM_{2.5}$ and $SO_2$ concentrations, widespread increase in $O_3MDA8$, and spatially variable trends in $NO_2$ concentrations, as reported previously (Silver et al., 2018). The model (Control simulation) simulates the widespread decline in $PM_{2.5}$ concentrations, with the median measured trend across China (-3.4 µg m$^{-3}$ year$^{-1}$) well simulated by the model (-3.5 µg m$^{-3}$ year$^{-1}$). In the measurements, 90% of significant trends are negative and 10% of significant trends are positive, with

positive trends mostly being in the Fenwei Plain region, Jiangxi and Anhui. No significant positive trends are simulated by the model, possibly due to coarse resolution of the model and the simplified scaling we apply to emissions for 2016 and 2017. WRF-Chem captures the widespread increase in $O_3MDA8$, but underestimates the magnitude of the trend by a factor 2 (2.7 µg m$^{-3}$ year$^{-1}$ in the measurements, versus 1.3 µg m$^{-3}$ year$^{-1}$ simulated by WRF-Chem). WRF-Chem simulates negative $O_3MDA8$ trends in the Sichuan Basin and Taiwan, whereas in the measured data, all regions have positive median trends.

The measurements show zero overall median trend in $NO_2$ concentrations, with 46% of sites with significant trends being negative and 54% positive. In contrast, WRF-Chem simulates widespread reductions in $NO_2$ concentrations, with 100% of significant sites exhibiting negative trends and a negative nationwide median trend of -2.2 µg m$^{-3}$ year$^{-1}$. The 7.0 % nationwide median decline in simulated $NO_2$ concentrations over 2015-2017, matches the 7.6 % decline in Chinese NOx emissions in the MEIC.

The measurements show a widespread decline in $SO_2$ concentrations, with a median nationwide trend of -1.9 µg m$^{-3}$ year$^{-1}$. WRF-Chem captures the direction of the trend, but the magnitude of the trend is overestimated by a factor 2. The 32.5 % decline in simulated nationwide median $SO_2$ concentrations over 2015-2017, matches the 37.8 % decline in $SO_2$ emissions in the MEIC.

## 3.2 Fixed emissions scenario

The model simulation where anthropogenic emissions in China were fixed at 2015 levels has a weak negative $PM_{2.5}$ trend (-0.6 µg m$^{-3}$ year$^{-1}$), a factor of six smaller than either the control simulation or the measurements (Figure 3). This suggests that the measured negative $PM_{2.5}$ trend has largely been driven by decreased anthropogenic emissions, with limited impact from interannual variability in meteorology. Chen et al. (2019) also concluded that emission reductions were the primary cause of reduced wintertime $PM_{2.5}$ across China during 2015-2017. Cheng et al., (2019) found that local and regional reductions in

anthropogenic emissions were the dominant cause of reduced $PM_{2.5}$ concentrations in Beijing between 2013 and 2017.
The median $O_3MDA8$ trend in the fixed emission simulation is 0.0 µg m$^{-3}$ year$^{-1}$. This suggests that interannual meteorological variation had little influence on $O_3$ trends at the China-wide scale during 2015-2017, which were largely driven by changing emissions. However, meteorological variability did drive regional changes in $O_3$. For example, in Guizhou province, a trend of -2.5 µg m$^{-3}$ year$^{-1}$ was calculated in the fixed emissions simulation. Li et al. (2019a) also report that the positive ozone trend

over 2013 to 2017 is due to changes in anthropogenic emissions, and the magnitude of their estimated trend of 1-3 ppbv year[-1] (approximately 2-6 µg m$^{-3}$ year$^{-1}$) is comparable to the 2.6 µg m$^{-3}$ year$^{-1}$ trend found in this study. Lu et al. (2019) analysed changes in $O_3$ between 2016 and 2017 and concluded that hotter and drier conditions in 2017 contributed to higher $O_3$ concentrations in that year. Liu and Wang (2020) reported a complex $O_3$ response during 2013 to 2017, with changing anthropogenic emission increasing $O_3$MDA8 in urban areas and decreasing it in rural areas, whereas meteorological changes

drove regionally contrasting changes in $O_3$MDA8 through changes in cloud cover, wind, and temperature and through driving changes in biogenic emissions.

    The fixed emission simulation also has a smaller $NO_2$ trend (-0.5 µg m$^{-3}$) compared to the control simulation (-2.2 µg m$^{-3}$ year$^{-1}$), demonstrating emission reductions that are estimated in the MEIC are also the main reason for the negative simulated $NO_2$ trend. However, unlike $PM_{2.5}$ and $O_3$, the $NO_2$ trend calculated from the fixed emission simulation more closely matches

measured trend. This may suggest that MEIC has overestimated the $NO_2$ emission reductions during 2015-2017. This suggestion is supported by recent satellite studies which found a slowing down or even reversal of $NO_2$ reductions during 2016-2019 (Li et al., 2019c), no significant trend in $NO_2$ during 2013-2017 (Huang et al., 2018), and increases in $NO_2$ concentration in the YRD, PRD and FWP regions during 2015-2017 (Feng et al., 2019). If $NO_x$ emissions decline too strongly in MEIC, this may contribute to the simulated underestimate of the positive observed $O_3$MDA8 trend in areas of China with a

$NO_x$ limited or mixed Ozone regimes that cover the majority of China (Jin and Holloway, 2015). Other work has suggested that increased $O_3$ concentrations are possibly linked to the rapid decline in aerosol (Li et al., 2019b). Liu and Wang (2020b) found that the reasons for increased $O_3$ concentrations during 2013-2017 were regionally dependent and that anthropogenic VOC emission reductions of 16-24% would have been needed to avoid increased concentrations.

    Table 2 compares the control and fixed emission simulations against $PM_{2.5}$, $O_3$ and $SO_2$ and $NO_2$ measurements in 2015, 2016

and 2017. In the control simulation model biases remain similar during 2015-2017. In the fixed emission simulation, model biases for $PM_{2.5}$, $O_3$ and $SO_2$ increase between 2015 and 2017. This further suggests that changing anthropogenic emissions during 2015-2017 have been the dominant cause of changing concentrations.

    An important future step is to understand how changing anthropogenic emissions, in terms of emission species or emission sectors, have contributed to observed trends in pollutant concentrations. Residential and industrial emissions are dominant

causes of $PM_{2.5}$ concentrations across much of China (Reddington et al., 2019), but it is not clear which emission sectors have contributed most to observed $PM_{2.5}$ trends. Cheng et al. (2019) suggests that emission controls in the residential and industrial sectors were the dominant causes for reduced $PM_{2.5}$ in Beijing between 2014 and 2017. Measurements of aerosol composition (Li et al., 2017b; Weagle et al., 2018) add confidence to model simulations and can inform our understanding of how aerosol chemistry responds to emission changes. However, except for Beijing, there is insufficient measurement data of how aerosol

composition has changed across China in recent years. Li et al. (2019a) found large declines in wintertime organics and sulfate and smaller declines in nitrate and ammonium in Beijing between 2014 and 2017. Zhou et al. (2019) also analysed aerosol composition data from Beijing and found large declines in all aerosol components except nitrate between 2011-12 and 2017-

18. Continuous measurements of aerosol composition across China are required to determine how different aerosol components are contributing to the observed $PM_{2.5}$ trend and to evaluate simulated responses to emission changes.

## 4 Health impacts of changes to PM$_{2.5}$ and O$_3$ concentrations

### 4.1 PM$_{2.5}$ health impacts

The control run simulated nation-wide population-weighted mean $PM_{2.5}$ concentration decreased by 12.8 % (10.1 µg m$^{-3}$), from 79.2 µg m$^{-3}$ in 2015 to 69.1 µg m$^{-3}$ in 2017. Greater decreases were simulated in more polluted and highly populated regions such as Beijing (-15.3 µg m$^{-3}$), Tianjin (-19.4 µg m$^{-3}$), Chongqing (province) (-14.2 µg m$^{-3}$) and Henan (-22.3 µg m$^{-3}$). Using the methodology of Burnett *et al.*, (2018), we estimate that mortality due to exposure to $PM_{2.5}$ decreased from 2 800 000 (CI: 2 299 000 – 3 302 000) premature mortalities in 2015, to 2 650 000 premature mortalities in 2017. The simulated reduction in $PM_{2.5}$ concentrations therefore reduced the number of premature mortalities attributable to $PM_{2.5}$ exposure by 150 000 (CI: 129 000 – 170 000) annual premature mortalities across China. The 12.8% reduction in $PM_{2.5}$ exposure only led to a 5% reduction in attributable mortality due to the non-linearity of the exposure-response function, which is less sensitive at higher exposure ranges (Conibear et al., 2018). The largest absolute reductions in premature mortality occur in Henan (15 000 deaths year$^{-1}$), Sichuan, Hebei and Tianjin (11 000 deaths year$^{-1}$) (Figure 4). The decline in $PM_{2.5}$ exposure also led to reduced morbidity with the Disability Adjusted Life Years (DALYs) rate per 100,000 population reduced from 159 to 150, with the largest changes occurring in central provinces such as (Supplementary Figure S3). Our results are comparable to Zheng *et al.*, (2017), who found that population weighted annual mean $PM_{2.5}$ concentrations decreased 21.5% during 2013 – 2015, resulting in a premature mortality decrease of 120 000 deaths year$^{-1}$. Ding et al., (2019) estimated that during 2013-2017, a nationwide $PM_{2.5}$ decrease of 9 µg m$^{-3}$ year$^{-1}$ caused premature mortalities pear year to decrease by 287 000, using the methodology from the GBD 2015 study, which estimates health impacts as having a weaker and less linear relationship to $PM_{2.5}$ concentrations. Yue et al. (2020) estimated that the annual number of mortalities in China attributable to $PM_{2.5}$ decreased by 64 000 (7%) from 2013 to 2017. Zhang et al. (2019) reported a 32% decline in population-weighted $PM_{2.5}$ concentration during 2013 to 2017, largely due to strengthened industrial emission standards and cleaner residential fuels.

### 4.2 O$_3$ health impacts

Increasing O$_3$ concentrations will result in an increase in health impacts that will act to offset some of the health benefits from declining $PM_{2.5}$ concentrations. WRF-Chem simulated O$_3$ concentrations across China during 2015-2017 to within 15% (NMB=-0.13), which is consistent with previous studies, but underestimated the magnitude of the observed O$_3$ trend. To provide an estimate of the health impacts due to exposure to O$_3$ we used simulated concentrations to estimate average exposure to O$_3$ over the 2015-2017 period. We estimate that exposure to O$_3$ caused an average of 143 000 (CI: 106 000 – 193 000) premature mortalities each year over 2015-2017. Applying the simulated change in O$_3$ concentrations would underestimate the change in exposure that has occurred, Instead, we estimated the impacts of increased O$_3$ by multiplying the average health

impacts over 2015-2017 by the measured relative change in $O_3MDA8$. Assuming linear behaviour, the 15% measured increase in $O_3MDA8$ would result in an increase of 21 000 premature mortalities per year. The exposure-outcome function is in reality sub-linear, so this is likely to be an overestimate. Regardless, this is substantially smaller than the 150 000 reduction in annual premature mortality due to reduced $PM_{2.5}$. We therefore suggest that changes in Chinese air pollution over 2015-2017 have likely had an overall beneficial impact on human health. The dominance of the $PM_{2.5}$ reduction over the $O_3$ increase on health impacts is also found in Dang and Liao (2019) who reported a 21% reduction in $PM_{2.5}$ and 12% increase in $O_3$ concentrations between 2012 and 2017 resulted in 268 000 fewer annual mortalities overall.

## 5 Conclusions

We used the WRF-Chem model to explore the drivers and impacts of changing air pollution across China during 2015-2017. A simulation with annually updated emissions was able to reproduce the measured negative trends in $PM_{2.5}$ concentrations over China during 2015 – 2017, while overestimating the negative trend in $SO_2$ and $NO_2$, and underestimating the positive trend in $O_3$. By comparing this with a simulation where emissions are held constant at 2015 levels, but meteorological forcing was updated, we show that interannual meteorological variation was not the main driver of the substantial trends in air pollutants that were observed across China during 2015 – 2017. Our work shows that reduced anthropogenic emissions are the main cause of reduced $PM_{2.5}$ concentrations across China, suggesting that the Chinese government's 'Air Pollution Prevention and Control Action Plan' has been effective at starting to control particulate pollution. We estimate that the 12.8% reduction in population-weighted $PM_{2.5}$ concentrations that occurred during 2015-2017 has reduced premature mortality due to exposure to $PM_{2.5}$ by 5.3%, preventing 150 000 premature mortalities across China annually. Despite these substantial reductions, $PM_{2.5}$ concentrations still exceed air quality guidelines and cause negative impacts on human health. We estimate that exposure to $O_3$ during 2015-2017 causes on average 143 000 premature mortalities across China each year. Increases in $O_3$ concentration over 2015-2017, may have increased this annual mortality by about 20 000 premature mortalities per year, substantially less than the reduction in premature mortality due to declining particulate pollution. Changes in air pollution across China during 2015-2017 are therefore likely to have led to overall positive benefits to human health, amounting to a ~5 % reduction of the ambient air pollution disease burden. However, to achieve larger reductions in the disease burden, further reductions in $PM_{2.5}$ concentrations are required, and pollution controls need to be designed that simultaneously reduce $PM_{2.5}$ and $O_3$ concentrations.

## Data availability

Data used to create all figures are available in the supplement. Air quality measurement data from mainland China's monitoring network is available from http://beijingair.sinaapp.com/. Air quality measurement data from Hong Kong is available from https://cd.epic.epd.gov.hk/EPICDI/air/station/. Air quality monitoring data from Taiwan is available from

https://taqm.epa.gov.tw/taqm/en/YearlyDataDownload.aspx. Data from all WRF-Chem model simulations and post-processing codes are available from the corresponding author on request.

**Author contributions**

BS, CLR, DVS and SRA designed the research. BS performed the WRF-Chem model simulations, analysed all the model data and wrote the manuscript. LC performed the health impact calculations. All authors contributed to scientific discussions and to the manuscript.

**Competing interests**

The authors declare that they have no conflict of interest.

**Acknowledgements**

We gratefully acknowledge the AIA Group Limited and Natural Environment Research Council (NE/S006680/1; NE/N006895/1) for funding for this research. Model simulations were undertaken on ARC3, part of the High Performance Computing facilities at the University of Leeds. We thank Qiang Zhang and Meng Li for providing MEIC data. We
acknowledge use of the WRF-Chem preprocessor tools bio_emiss, anthro_emiss, fire_emiss, and mozbc provided by the Atmospheric Chemistry Observations and Modeling Lab (ACOM) of the NCAR. We acknowledge use of NCAR/ACOM .MOZART-4 global model output available at http://www.acom.ucar.edu/wrf-chem/mozart.shtml (last accessed 12[th] December 2018). We thank the SPARTAN project for its effort in establishing and maintaining the site in Beijing. The SPARTAN network was initiated with funding from the Natural Sciences and Engineering Research Council of Canada.

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

**Table 1. Chinese pollutant emissions (Tg yr⁻¹) during 2015 to 2017 from MEIC.**

| | $SO_2$ | $NO_x$ | NMVOC | $NH_3$ | CO | TSP | $PM_{10}$ | $PM_{2.5}$ | BC | OC | $CO_2$ |
|------|------|------|-------|------|-------|------|------|------|-----|-----|---------|
| 2015 | 16.9 | 23.7 | 28.6 | 10.5 | 153.6 | 21.9 | 12.3 | 9.1 | 1.4 | 2.5 | 10347.2 |
| 2016 | 13.4 | 22.5 | 28.4 | 10.2 | 142 | 17.9 | 10.8 | 8.1 | 1.3 | 2.3 | 10290.7 |
| 2017 | 10.5 | 21.9 | 28.6 | 10.2 | 136.2 | 16.7 | 10.2 | 7.6 | 1.2 | 2.1 | 10434.3 |


**Table 2.  Model evaluation shown as a normalised mean bias (NMB). Evaluation is shown separately for the control and fixed emission simulations. The NMB for 2015-2017 is compared to individual years.**

|  | $PM_{2.5}$ | $O_3$ | $NO_2$ | $SO_2$ |
|---|---|---|---|---|
| Control |  |  |  |  |
| 2015-2017 | 0.49 | -0.15 | 1.2 | 0.09 |
| 2015 | 0.5 | -0.12 | 1.32 | 0.17 |
| 2016 | 0.47 | -0.14 | 1.20 | 0.05 |
| 2017 | 0.5 | -0.21 | 1.10 | 0.04 |
| Fixed emissions |  |  |  |  |
| 2015-2017 | 0.57 | -0.18 | 1.26 | 0.35 |
| 2015 | 0.50 | -0.12 | 1.32 | 0.17 |
| 2016 | 0.56 | -0.16 | 1.28 | 0.31 |
| 2017 | 0.66 | -0.24 | 1.20 | 0.65 |

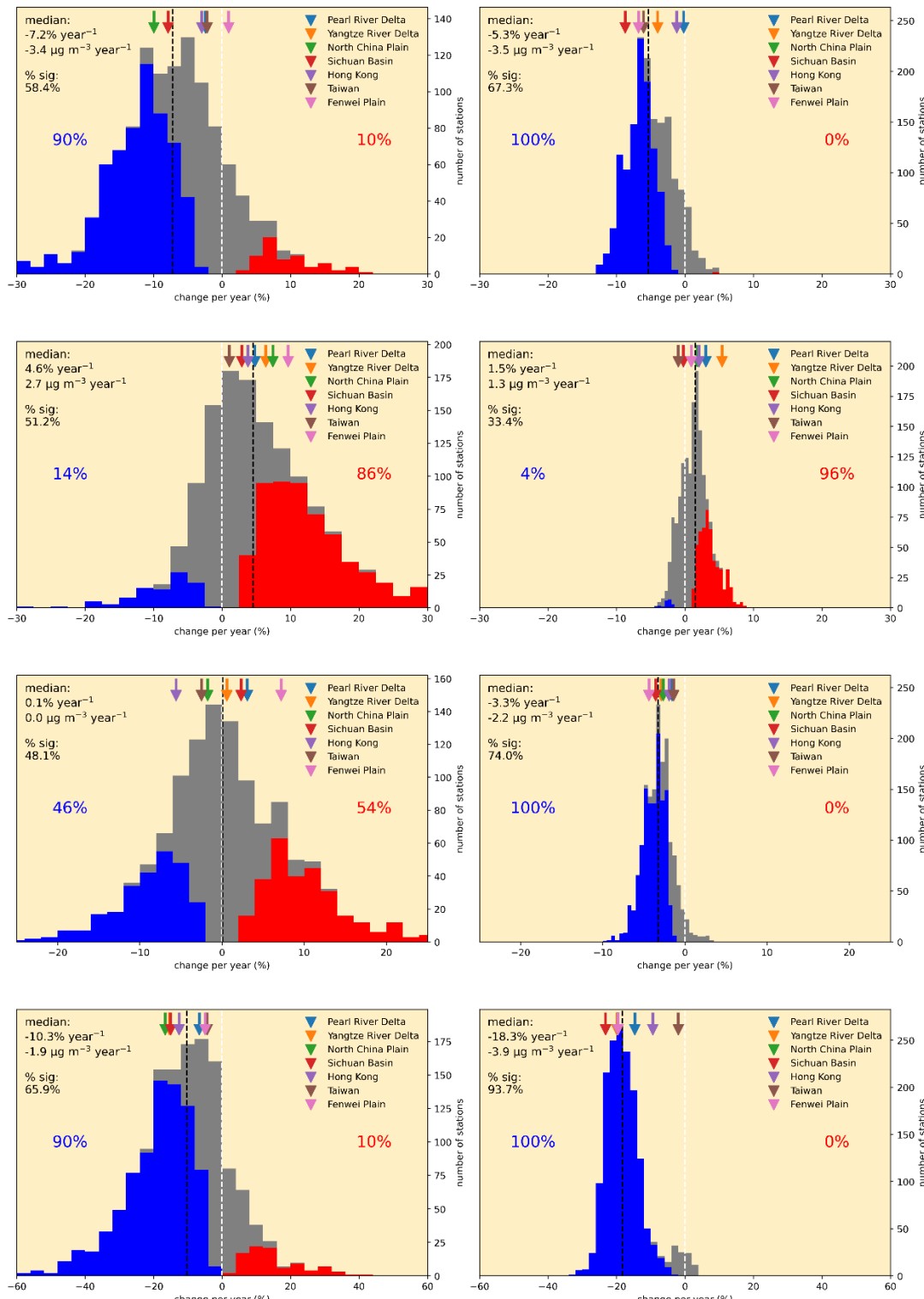

**Figure 1: Histograms showing the frequency distribution of trends in concentrations of (a,b) PM$_{2.5}$, (c,d) O$_3$MDA8, (e,f) NO$_2$, (g,h) SO$_2$ across China and Taiwan during 2015–2017. Measured trends (left hand panels) are compared to simulated trends (right hand panels). The median relative and absolute trend as well as the percentage of stations with significant trends is shown on each panel. The percentage of significant trends that are negative (blue) or positive (red) are also shown. The black dotted line shows the median trend across all sites, while the white dotted line shows zero. Arrows show the median trend for the regional domain: Pearl River**
**Delta (PRD), Yangtze River Delta (YRD), North China Plain (NCP), Sichuan Basin (SCB), Hong Kong (HK), Taiwan (TW) and the Fenwei Plain (FWP).**

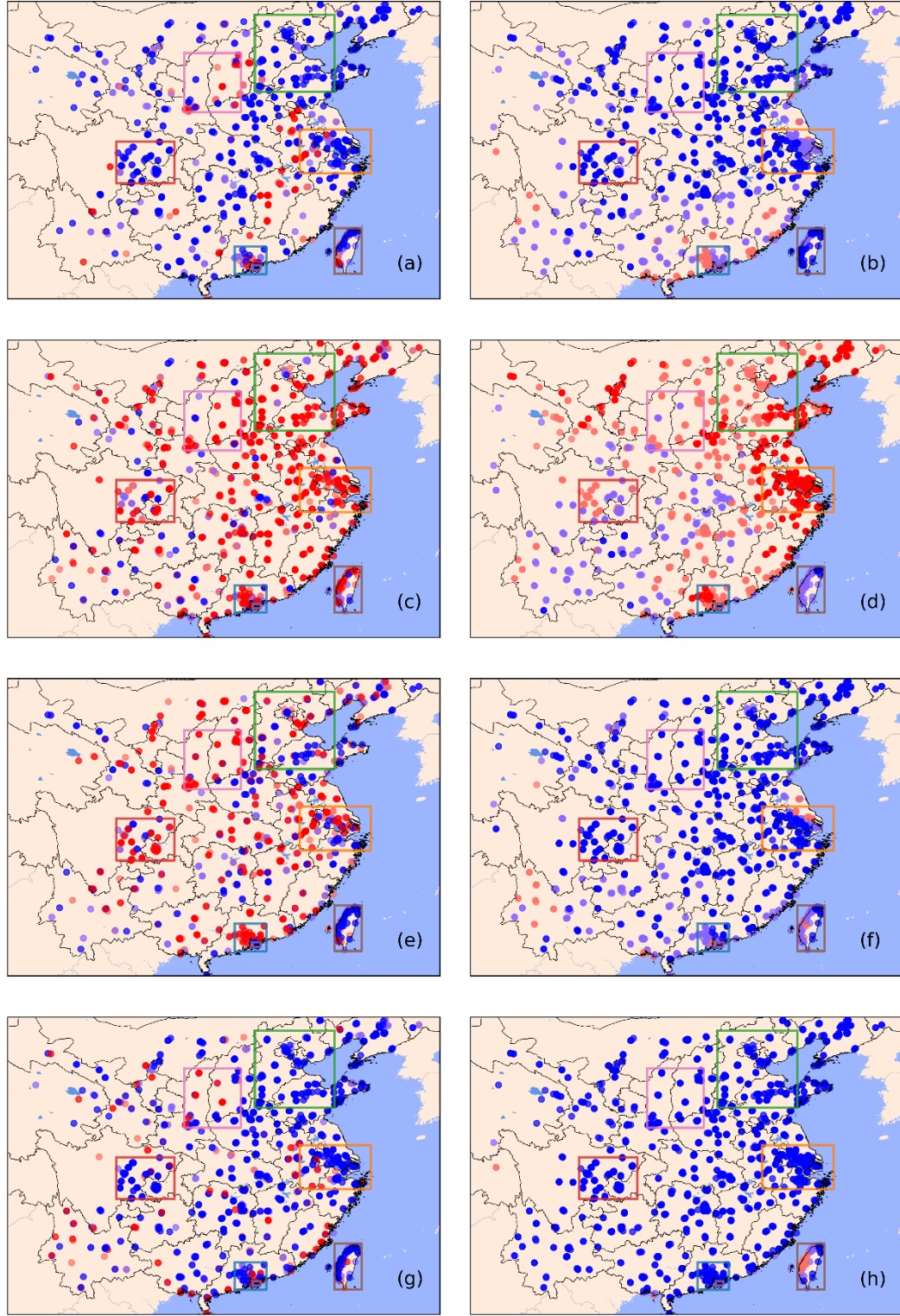

 **Figure 2. Map showing the spatial distribution of trends in concentrations of (a,b) PM$_{2.5}$, (c,d) O$_3$MDA8, (e,f) NO$_2$, (g,h) SO$_2$ across China and Taiwan during 2015–2017. Measured trends (left hand panels) are compared to simulated trends (right hand panels). Red indicates a significant positive trend, whereas blue indicates a significant negative trend. Lighted coloured circles indicated a statistically insignificant trend. Coloured boxes show the regional domains: Pearl River Delta (blue), Yangtze River Delta (orange), North China Plain (green), Sichuan Basin (red), Hong Kong (purple), Taiwan (brown) and the Fenwei Plain (pink).**

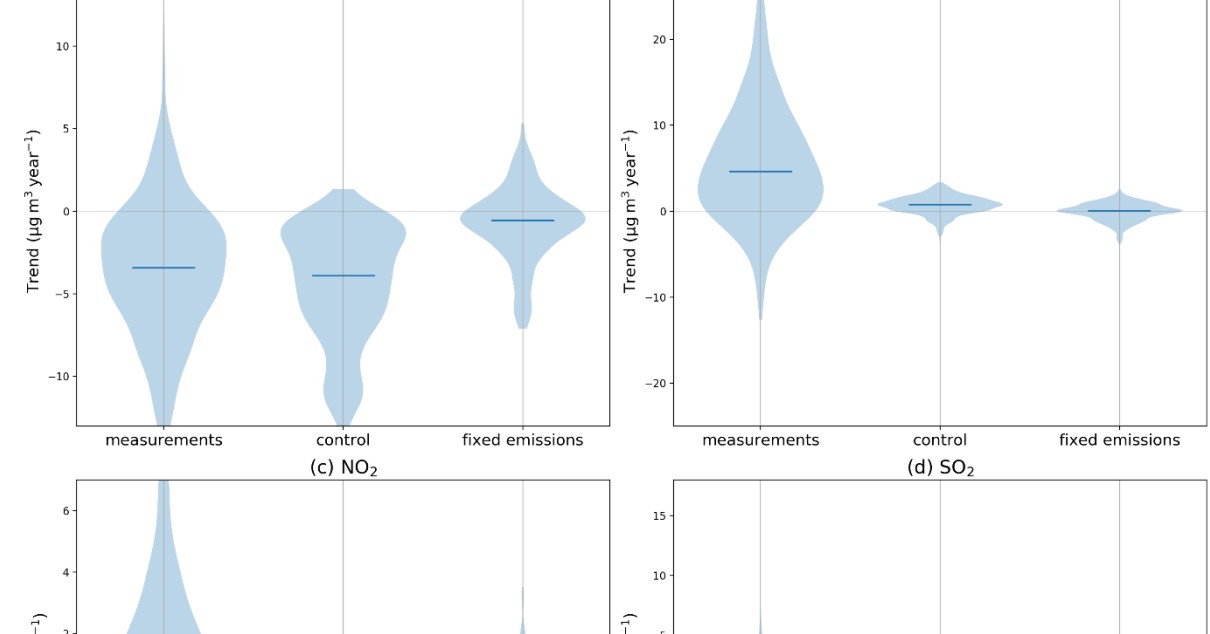

**Figure 3. Comparison of measured and simulated concentration trends during 2015 to 2017. The left violin shows the measured trend, the centre shows the simulated trend with varying emissions and meteorology (control), and the right shows the simulated trends for the fixed emissions simulation. a) PM$_{2.5}$, b) O$_3$MDA8 , c) NO$_2$, d) SO$_2$. The solid line shows the median absolute trend, and**
 **the shaded area shows a smoothed relative frequency distribution.**

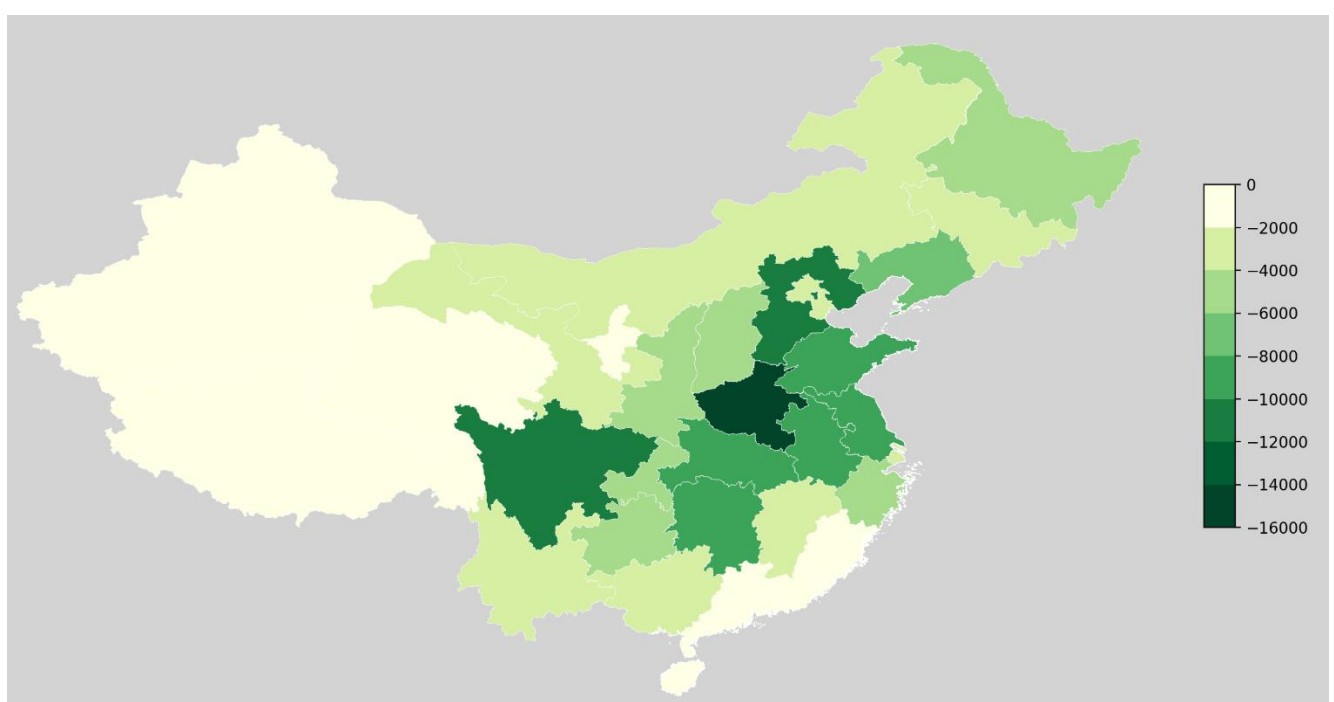

**Figure 4. Simulated change during 2015-2017 in annual premature mortality per year due to changes in exposure to ambient PM2.5. Results are shown at the province scale.**