# Peer review of "Pollutant emission reductions deliver decreased PM2.5-caused mortality across China during 2015-2017"

_Atmospheric Chemistry and Physics, 2019_

## Referee Comment (RC1) · Anonymous Referee #2 · 13 Mar 2020

The manuscript presents a study on estimating the changes of mortality due to air-pollutants in China in 2015-2017 and explaining the causes of it using WRF-Chem simulations. For this, modeled trends are compared with observed ones to provide reliability in the model estimates. This study represents good contributions to the field and it's within the scope of ACP. I think the paper needs a bit more work before it's ready for publication based on the comments below.

My main comment is the following. Given the issues in the modeled trends of the PM2.5 precursors (SO2, NOx), getting the right trend for PM2.5 could be do to a cancellation of errors, so you might be getting the right trend for the wrong reasons. I would like to

[Figure]

encourage the authors to look into more details on this topic. For instance, analyze the model results by aerosol composition and how are the trends of each specie to assess the role of each of them. I would also encourage the authors to collaborate with other researchers that maintain sites where this speciation is observed and so the speciated comparison can be done as well. An example is the Beijing site from the Spartan network (https://www.spartan-network.org/beijing-china), but I'm sure there are many more. Even if a few sites are included this could provide useful information.

Comments by line:

66-70. Please list some references on the second approach.

79. Please briefly summarize the quality control process

Section 2.2. Any previous work where you have used this or similar configuration with positive results in terms of meteorology, PM2.5 and O3? In this work you are not much model evaluation other than the evaluation of the trends and brief statistics in sections 2.3. Adding evaluation on the ability of this model configuration to capture aerosol speciation would also be desirable.

103 Hodzic and Jimenez, and Knote et al. papers described two very different SOA schemes, please specify which one you are using.

Section 2.4. Can you briefly describe the exposure response functions used for PM2.5 and O3?

129. After reading section 3.1, I don't think the trends compare largely well as stated in this sentence. You could make this point for PM2.5, but for the others, although the sign is generally correct, the magnitudes tends to be off by at least factor of 2. For the case of NO2 the sign of the trend is not even well captured. Please revise to better represent the actual results

129-136. Can you add additional analysis in whether the model captures the regions with more negative (and more positive) trends? I see this info in the plots but it's not

discussed

150. Is clear to me that natural emissions remained equal in the two emission scenarios, but what did the authors do for biomass burning emissions?

153-154. Chen et al. (2019) also found that there were periods where meteorology did play a role, can you compare your results to theirs?

155-156. I would say "little influence" rather than "no influence" as you are basing your analysis in a model that contains uncertainties.

166-167. This statement depends if the region is NOx or VOC limited. Might be good to include these indicators from the model perspective to shows that this is what's actually happening in the model.

168. The Li et al. (2019a) study blames heterogeneous chemistry happening in declining particles for the negative trend. Is this process included in this WRF-Chem configuration and how this influences your results? An attempt to compute similar metrics as in the Li study might be good to intercompare results.

191-196. Is not clear in this paragraph where you consider the model issues on O3 trends.

Minor Edits

75. I believe ACP policy is to not use links but references, please check.

124. Fix issue with the symbol after 0.05

329. Should this be 2019b?

---

## Referee Comment (RC2) · Anonymous Referee #1 · 17 Mar 2020

Summary:

In this work, the authors use a chemical transport model (WRF-Chem) to demonstrate that emission controls rather than meteorology have been driving the air-quality improvement in China in recent years. Additionally, the authors calculate the number of lives saved from China's 'Air Pollution Prevention and Control Action Plan' between 2015 and 2017. This manuscript is of good scientific and presentation quality and in a highly-relevant area of research. However, there have already been several articles published on (1) whether meteorology or emissions are driving air quality changes in China and (2) the health impacts of the stringent emissions controls in China. Overall, a

better case for the novelty of this work needs to be made in the motivation/introduction of the paper (see overall comment below). Additionally, there are several places where more detail and/or discussion is needed (see Line-by-line comments below).

General Comments:

There have already been several articles evaluating the impacts of meteorology vs emissions on changes of PM2.5 in China (e.g., https://doi.org/10.5194/acp-19-7409-2019, https://doi.org/10.1289/EHP4157) and several papers that calculated the health effects of the stringent emissions controls in China in recent years (e.g., https://doi.org/10.1289/EHP4157, https://doi.org/10.1088/1748-9326/aa8a32). Many of these papers were mentioned in the results/discussion section of this work. In the introduction, the authors should mention some of this closely-related previous work and discuss what distinguishes this work from previous studies.

Specific Comments:

Line 75: What was done to clean the dataset? Please provide more information.

Line 106-108: Suggest providing context for these NMB. For example, how do they compare to previous work? How will they impact air pollution-mortality estimates? Are these NMB calculated using the "control" simulation? Is the NMB greater if the measurements are compared with the "fixed emissions" simulation?

Line 109-112: What was the measurement/simulation bias for each year? If it does not change substantially, this would help validate the methodology used here for decoupling the impacts of emissions and meteorology on PM2.5 and O3 levels.

Line 113-117: Please provide more information here. What is meant by "interpolated model data"? Did the authors look only at the model estimates that coincided with the measurements? Please provide some information about the method that was used for the measurements data, so the reader doesn't need to look at the Silver et al, 2018, unless they are interested in a high level of detail of the methods. What method was

used to deseasonalize the data?

Line 125: It's useful to calculate the changes in mortality based on exposure alone, but I also suggest calculating the number of PM2.5 and O3 mortalities with population/age/baseline mortality data from 2017 to provide more realistic mortality estimates for 2017. It would be useful to see if the air pollution reductions in 2017 have increased benefits due to the increased population from 2015, for example.

Line 153: How did the Chen et al, 2019 trends compare and how did their emissions scaling compare?

Line 157: Please provide more information on Guizhou and Li et al, 2018. Why do they see different trends? Did they use different emissions scaling? Did they look at different regions of China?

Line 190-191: Suggest that the authors provide more context for what is meant by a "reasonable" NMB.

Figure 2: Which color represents which region mentioned in the figure caption.

Figure 2: Suggest making the dots that don't have significant trends (i.e., the gray dots) a lighter blue or red color. Even if the trends are statistically significant the direction of the trend will still provide information.

Figure 4: I find this color scale to interpret because the mid-range yellows and greens all look similar. I suggest binning the color scale to make it easier to see how many lives are saved in each province.

Technical Corrections:

Line 157: Missing year of Guizhou. I also couldn't find this reference in the reference list.

Line 185: Should be "per year".

Line 247: The first two papers in the reference section don't seem to be in alphabetical order.

---

## Author Comment (AC1) · 9 Jul 2020

**Response to Reviewer Comments on 'Pollutant emission reductions deliver decreased PM$_{2.5}$-caused mortality across China during 2015–2017'**

We thank the reviewers for their positive and constructive comments on our manuscript. We have addressed all of the referee comments through changes to our manuscript. We have also updated our manuscript to include references to recent relevant studies. We feel our revised manuscript is improved and we hope it is acceptable for publication. In the below response, referee comments are in italics and our response is in plain text. Line numbers refers to the revised manuscript.

**Response to Anonymous Referee #1**

*Summary:*

*In this work, the authors use a chemical transport model (WRF-Chem) to demonstrate that emission controls rather than meteorology have been driving the air-quality improvement in China in recent years. Additionally, the authors calculate the number of lives saved from China's 'Air Pollution Prevention and Control Action Plan' between 2015 and 2017. This manuscript is of good scientific and presentation quality and in a highly-relevant area of research. However, there have already been several articles published on (1) whether meteorology or emissions are driving air quality changes in China and (2) the health impacts of the stringent emissions controls in China. Overall, a better case for the novelty of this work needs to be made in the motivation/introduction of the paper (see overall comment below). Additionally, there are several places where more detail and/or discussion is needed (see Line-by-line comments below).*

*General Comments:*

*There have already been several articles evaluating the impacts of meteorology vs emissions on changes of PM2.5 in China (e.g., https://doi.org/10.5194/acp-19- 7409-2019, https://doi.org/10.1289/EHP4157) and several papers that calculated the health effects of the stringent emissions controls in China in recent years (e.g., https://doi.org/10.1289/EHP4157, https://doi.org/10.1088/1748-9326/aa8a32). Many of these papers were mentioned in the results/discussion section of this work. In the introduction, the authors should mention some of this closely-related previous work and discuss what distinguishes this work from previous studies.*

We thank the reviewer for their comment. We have added a paragraph (on lines 69-74) which mentions similar papers, and discusses the distinguishing features of our study:

"There are a limited number of modelling studies that attempt to separate the influence of meteorology and emissions changes on recent air quality trends in China. Chen et al. (2019) use WRF-Chem with 2010 emissions to examine the drivers of trends in wintertime PM. Ding et al. (2019) use WRF-CMAQ to evaluate importance of emissions, meteorology and demographic changes on PM$_{2.5}$ related mortality during 2013-2017. Our paper adds to these previous studies by evaluating the ability of WRF-Chem to simulate trends in NO$_2$, O$_3$ and SO$_2$ as well as PM, using the most recent emissions and evaluated against a comprehensive measurement dataset."

*Specific Comments:*

*Line 75: What was done to clean the dataset? Please provide more information.*

We already included some brief details in our manuscript and a full description is given in Silver et al. (2018). As suggested by the referee we provide more information (now on line 85-87) to further describe the data cleaning procedure, through the text:

'We conducted quality control on the measured data following the methods outlined in Silver et al. (2018), which include excluding data with a high proportion of repeated measurements and periods of low variability'.

*Line 106-108: Suggest providing context for these NMB. For example, how do they compare to previous work? How will they impact air pollution-mortality estimates?*

We have added a table of NMB statistics from previous studies to the supplement (Table S1). We included studies that also use WRF-Chem in China, and have a similar model setup.

|  | $PM_{2.5}$ | $PM_{10}$ | $O_3$ | $NO_2$ | $SO_2$ | CO |
|---|---|---|---|---|---|---|
| Zhang et al. (2016) (Hong Kong) |  | -0.47 to -0.07 | 0.88 to 1.6 | -0.88 to -0.83 | -0.84 to -0.59 | -0.72 to -0.55 |
| Zhang et al. (2016) (China) |  | -0.38 to -0.03 |  |  | -0.8 to -0.72 |  |
| (Wang et al., 2016) (N China, January) | 0.28 to 0.47 | 0.00 to 0.08 |  | 0.09 to 0.27 | 0.33 to 0.91 | 0.01 to 0.12 |
| Zhou et al. (2017) (forecast) |  | -0.36 |  | -0.05 | -0.18 | -0.4 |
| This paper | 0.49 | -0.09 | -0.15 | 1.2 | 0.09 | -0.35 |

We add the following text to the main paper (Line 117-118):

"Model biases were similar to previous model studies in China (Supplementary Table 1)."

*Are these NMB calculated using the "control" simulation? Is the NMB greater if the measurements are compared with the "fixed emissions" simulation?*

This is a good suggestion. Yes, the NMB is further from zero in the fixed emissions scenario. We include the table below in the main paper (Table 2):

|  | PM2.5 | O3 | NO2 | SO2 |
|---|---|---|---|---|
| Control |  |  |  |  |
| 2015-2017 | 0.49 | -0.15 | 1.2 | 0.09 |
| 2015 | 0.5 | -0.12 | 1.32 | 0.17 |
| 2016 | 0.47 | -0.14 | 1.20 | 0.05 |
| 2017 | 0.5 | -0.21 | 1.10 | 0.04 |
| Fixed emissions |  |  |  |  |
| 2015-2017 | 0.57 | -0.18 | 1.26 | 0.35 |
| 2015 | 0.50 | -0.12 | 1.32 | 0.17 |
| 2016 | 0.56 | -0.16 | 1.28 | 0.31 |
| 2017 | 0.66 | -0.24 | 1.20 | 0.65 |

**Table 2. Model evaluation shown as a normalised mean bias (NMB). Evaluation is shown separately for the control and fixed emission simulations. The NMB for 2015-2017 is compared to individual years.**

*Line 109-112: What was the measurement/simulation bias for each year? If it does not change substantially, this would help validate the methodology used here for decoupling the impacts of emissions and meteorology on PM2.5 and O3 levels.*

This was a useful suggestion, and the information has been included in Table 2 (see above), which has been added to the manuscript. We also added these sentences (lines 204-207):

"Table 2 compares the control and fixed emission simulations against $PM_{2.5}$, $O_3$ and $SO_2$ and $NO_2$ measurements in 2015, 2016 and 2017. In the control simulation model biases remain similar during 2015-2017. In the fixed emission simulation, model biases for $PM_{2.5}$, $O_3$ and $SO_2$ increase between 2015 and 2017. This further suggests that changing anthropogenic emissions during 2015-2017 have been the dominant cause of changing concentrations."

*Line 113-117: Please provide more information here. What is meant by "interpolated model data"? Did the authors look only at the model estimates that coincided with the measurements? Please provide some information about the method that was used for the measurements data, so the reader doesn't need to look at the Silver et al, 2018, unless they are interested in a high level of detail of the methods. What method was used to deseasonalize the data?*

We explain what we mean by interpolated model data on line 115:

"For comparison with the measurements, we sampled the model at the station locations using linear interpolation."

We have amended the text (lines 134-135) to explain how we deseasonalise the data. For clarity, we remove mention of interpolated data here

"Trends in the model data were calculated using the same method as the measurement data (Silver et al., 2018). The hourly data are averaged to monthly means, which are then deseasonalised using locally weighted scatterplot smoothing."

When evaluating model trends (Section 3) we match the model to the measurements. When exploring changes in exposure and impacts on health (Section 4) we calculate population-weighted exposure.

*Line 125: It's useful to calculate the changes in mortality based on exposure alone, but I also suggest calculating the number of PM2.5 and O3 mortalities with population/age/baseline mortality data from 2017 to provide more realistic mortality estimates for 2017. It would be useful to see if the air pollution reductions in 2017 have increased benefits due to the increased population from 2015, for example.*

We thank the reviewer for their suggestion and agree that it can be interesting to compare the effects of exposure and demography air pollution health impact. However, in the context of this study, which focusses on distinguishing the emissions and meteorological effects on air quality, we believe that including mortality changes due to changing age distribution, population and baseline mortality is outside the scope of the current study. We isolated the health impacts of the change in exposure by keeping the time frame constant thereby removing the influence of the confounding variations in population ageing, population size, and baseline disease levels. As the reviewer points out, health impact assessments are sensitive to the underlying epidemiological data and have rapidly developing methodologies. We now explain our approach more clearly (Line 150-152):

"Health impacts depend on population count, population age, and baseline mortality rates which have changed over the period studied (Butt et al., 2017). To isolate the impacts of changing air pollution, other variables were kept constant for 2015-2017."

*Line 153: How did the Chen et al, 2019 trends compare and how did their emissions scaling compare?*

Chen et al. (2019) did not scale emissions. They use 2010 emissions, which led to a 'severe overestimation' in PM, due to the decline in emissions that has occurred since 2010. Chen et al. (2019) focused on wintertime (January). They also suggested that reductions in emissions had contributed to reduced wintertime $PM_{2.5}$ concentrations (see Line 178-179):

"Chen et al. (2019) also concluded that emission reductions were the primary cause of reduced wintertime $PM_{2.5}$ across China during 2015-2017."

*Line 157: Please provide more information on Guizhou and Li et al, 2018. Why do they see different trends? Did they use different emissions scaling? Did they look at different regions of China?*

The reference to "Guizhou" is to the trend in the province of Guizhou in the fixed emissions run. The text has been amended to "Guizhou province" make this clear. For Li et al (2019a) [corrected], their trend estimate has been added in lines 184-186:

"Li et al. (2019a) also report that the positive ozone trend over 2013 to 2017 is due to changes in anthropogenic emissions, and the magnitude of their estimated trend of 1-3 ppbv year$^{-1}$ (approximately 2-6 µg m$^{-3}$ year$^{-1}$)  is comparable to the 2.6 µg m$^{-3}$ year$^{-1}$ trend found in this study."

*Line 190-191: Suggest that the authors provide more context for what is meant by a "reasonable" NMB.*

This word reasonable has been removed to remove subjectivity around the NMB. Additionally, a table has been added to the supplement (Table S1) that contains NMB recorded in other papers that have a similar WRF-Chem model setup.

*Figure 2: Which color represents which region mentioned in the figure caption.*

This has been added to the figure caption.

*Figure 2: Suggest making the dots that don't have significant trends (i.e., the gray dots) a lighter blue or red color. Even if the trends are statistically significant the direction of the trend will still provide information.*

Thank you for your suggestion, the figure caption has been updated to include the colour for each region, which corresponds to the legend in Figure 1.

*Figure 4: I find this color scale to interpret because the mid-range yellows and greens all look similar. I suggest binning the color scale to make it easier to see how many lives are saved in each province.*

Thank you for your suggestion, this has been changed to a binned colour scale.

*Technical Corrections:*

*Line 157: Missing year of Guizhou. I also couldn't find this reference in the reference list.*

Guizhou refers to the province of Guizhou, the text has been amended to make this clearer

*Line 185: Should be "per year".*

This has been corrected

*Line 247: The first two papers in the reference section don't seem to be in alphabetical order*

These papers are ordered by "A" of "van der A". We will check with the journal editorial team to ensure we are following journal guidelines.

**Response to Anonymous Referee #2**

*The manuscript presents a study on estimating the changes of mortality due to air pollutants in China in 2015-2017 and explaining the causes of it using WRF-Chem simulations. For this, modeled trends are compared with observed ones to provide reliability in the model estimates. This study represents good contributions to the field and it's within the scope of ACP. I think the paper needs a bit more work before it's ready for publication based on the comments below. My main comment is the following. Given the issues in the modelled trends of the PM2.5 precursors (SO2, NOx), getting the right trend for PM2.5 could be do to a cancellation of errors, so you might be getting the right trend for the wrong reasons. I would like to encourage the authors to look into more details on this topic. For instance, analyze the model results by aerosol composition and how are the trends of each specie to assess the role of each of them. I would also encourage the authors to collaborate with other researchers that maintain sites where this speciation is observed and so the speciated comparison can be done as well. An example is the Beijing site from the Spartan network (https://www.spartan-network.org/beijing-china), but I'm sure there are many more. Even if a few sites are included this could provide useful information*

We thank the reviewer for their comment, and agree that it is useful to evaluate simulated aerosol speciation by comparison with measurements. Unfortunately there is insufficient data to carefully evaluate the trend in aerosol speciation across China during the period analysed.

As suggested, we compare the model with the Beijing SPARTAN data (Snider et al., 2015, 2016). The model underestimates sulfate and overestimates nitrate at this particular site. The results of the comparison have been added to the supplement (Figure S4). We also compare to another set of measurements from Beijing that were reported by Zhou et al. (2019) (Figure S5), which point to an underestimate of sulfate, nitrate and ammonium in winter, but reasonable estimations in other seasons.

To extend the evaluation beyond Beijing, we also use a dataset of speciated aerosol data from different field campaigns compiled by Li et al. (2017) (Figure S6). Data spans multiple years from 2006-2013, so the comparison is complicated by comparing the model for 2015 against years of different meteorology and emissions. For this reason, we only compare means values across the campaign which will reduce the impacts of different meteorological conditions between the measurement and model. Nevertheless, this comparison also suggests the model underestimates sulfate and overestimate nitrate. The large changes in emissions (in particular the large decline in $SO_2$ emissions) over this period is likely to cause at least some of the underestimate in sulfate. The results of this comparison are added to the Supplement (Fig. S6). We add the following text to the paper on lines (118-128):

"We also evaluated the model against speciated aerosol measurements from the Surface PARTiculate mAtter Network (SPARTAN) (Snider et al., 2015, 2016) site in Beijing (https://www.spartan-network.org/beijing-china, last accessed: 2nd July 2020) (Fig S4),  Aerosol Chemical Speciation Monitor measurements from Beijing (Zhou et al. (2019)) (Figure S5) and Aerosol

Mass Spectrometer measurements from across China (Li et al., 2017) (Fig S6). Measurements reported by Li et al. (2017b) were made from various years spanning 2006 to 2013 and do not match the years simulated by the model. Comparison against these data show that the model underestimates the sulfate fraction in $PM_{2.5}$, while overestimating the nitrate fraction. Underestimation of sulfate in comparison to Li et al., (2017b) will partly be caused by the large decline in $SO_2$ emissions that has occurred in the last decade (Zheng et al., 2018). Underestimate of sulfate, particularly in winter, and overestimation of nitrate are consistent with previous modelling studies (Shao et al., 2019) including those using WRF-chem (Zhou et al., 2019). Newly proposed mechanisms to explain the rapid sulfate formation in China's winter haze (Gen et al., 2019; Shao et al., 2019; Xue et al., 2014; Zhang et al., 2019) need to be included and evaluated in models."

There is insufficient data to evaluate the trends in aerosol speciation over the 2015 to 2017 period. We add the following text to the paper (Line 208-19):

"An important future step is to understand how changing anthropogenic emissions, in terms of emission species or emission sectors, have contributed to observed trends in pollutant concentrations. Residential and industrial emissions are dominant causes of PM2.5 concentrations across much of China (Reddington et al., 2019), but it is not clear which emission sectors have contributed most to observed $PM_{2.5}$ trends. Cheng et al. (2019) suggests that emission controls in the residential and industrial sectors were the dominant causes for reduced $PM_{2.5}$ in Beijing between 2014 and 2017. Measurements of aerosol composition (Li et al., 2017; Weagle et al., 2018) add confidence to model simulations and can inform our understanding of how aerosol chemistry responds to emission changes. However, except for Beijing, there is insufficient measurement data of how aerosol composition has changed across China in recent years. Li et al. (2019a) found large declines in wintertime organics and sulfate and smaller declines in nitrate and ammonium in Beijing between 2014 and 2017. Zhou et al. (2019) also analysed aerosol composition data from Beijing and found large declines in all aerosol components except nitrate between 2011-12 and 2017-18. Continuous measurements of aerosol composition across China are required to determine how different aerosol components are contributing to the observed $PM_{2.5}$ trend and to evaluate simulated responses to emission changes."

*Comments by line:*

*66-70. Please list some references on the second approach.*

Two references, Ansari et al. (2019) and Xing et al. (2011) have been added as examples of this approach on line 68.

*79. Please briefly summarize the quality control process*

A brief summary of the quality control process has been added on lines 86-87: '…which include excluding data with a high proportion of repeated measurements and periods of low variability'

*Section 2.2. Any previous work where you have used this or similar configuration with positive results in terms of meteorology, PM2.5 and O3? In this work you are not much model evaluation other than the evaluation of the trends and brief statistics in sections*

Further detail of the model evaluation has been added in Table 2 (see comments to Referee #1). Previous studies using a similar model set-up that demonstrate the model's ability to capture PM include Reddington et al. (2019). A table (Table S1) has been added to the supplementary showing the comparable NMB statistics of published research that use a similar WRF-Chem setup to simulate air pollution in China.

*2.3. Adding evaluation on the ability of this model configuration to capture aerosol speciation would also be desirable.*

Two plots that show the ability of the model to capture aerosol speciation, Figures S4 and S5, have been added to the supplement. We have also added text to the manuscript (see response to Referee #1).

*103 Hodzic and Jimenez, and Knote et al. papers described two very different SOA schemes, please specify which one you are using.*

We use the scheme described in Hodzic and Jimenez. The second reference on line 112 has been corrected to Hodzic and Knote (2014).

*Section 2.4. Can you briefly describe the exposure response functions used for PM2.5 and O3?*

We have added more information to the paper on the shape of the exposure-response functions. For $PM_{2.5}$ on lines 140-142:

"Health impacts are estimated for ambient $PM_{2.5}$ using the Global Exposure Mortality Model (GEMM) (Burnett et al., 2018), which uses cohort studies to estimate health risks integrated over a range of $PM_{2.5}$ concentrations. GEMM applies a supralinear association between exposure and risk at lower concentrations and then a near-linear association at higher concentrations."

And for $O_3$ on lines 144-148:

For ambient $O_3$, we used the methodology of the Global Burden of Disease (GBD) study for 2017 (GBD 2017 Risk Factor Collaborators et al., 2018) to estimate the mortality caused by chronic obstructive pulmonary disease, which is based on exposure and risk information from five epidemiological cohorts. It estimates a near-linear relationship between exposure and risk at lower concentrations of $O_3$, and a sub-linear association at higher concentrations.

*129. After reading section 3.1, I don't think the trends compare largely well as stated in this sentence. You could make this point for PM2.5, but for the others, although the sign is generally correct, the magnitudes tends to be off by at least factor of 2. For the case of NO2 the sign of the trend is not even well captured. Please revise to better represent the actual results*

We have changed this sentence to remove the statement that all the trends compare well. The sentence now reads (line 155):

"Figure 1 and 2 compare measured and simulated air quality trends over China during 2015 to 2017"

*129-136. Can you add additional analysis in whether the model captures the regions with more negative (and more positive) trends? I see this info in the plots but it's not discussed.*

The regional distribution of trends for PM are discussed in lines 160-161 and for $O_3$ in lines 163-164. Line 166-167 mentions the fact that the model does not capture the regional distribution of trends for $NO_2$.

*150. Is clear to me that natural emissions remained equal in the two emission scenarios, but what did the authors do for biomass burning emissions?*

Biomass burning emissions are from the FINN inventory in both scenarios. We now clarify this in the manuscript (Line 132-133):

"Both simulations include interannual variability in biogenic and biomass burning emissions, allowing us to isolate the impacts of changing anthropogenic emissions."

*153-154. Chen et al. (2019) also found that there were periods where meteorology did play a role, can you compare your results to theirs?*

We add the following sentence to our manuscript (line 178-179):

Chen et al. (2019) also concluded that emission reductions were the primary cause of reduced wintertime $PM_{2.5}$ across China during 2015-2017

*155-156. I would say "little influence" rather than "no influence" as you are basing your analysis in a model that contains uncertainties.*

This has been corrected in the text.

*166-167. This statement depends if the region is NOx or VOC limited. Might be good to include these indicators from the model perspective to shows that this is what's actually happening in the model.*

We agree with the reviewer that this statement depends on the ozone regime, and we have added a reference to a satellite study which finds that a $NO_x$ limited or mixed regime dominates across China. We amended the sentence (lines 198-200):

"If $NO_x$ emissions decline too strongly in MEIC, this may contribute to the simulated underestimate of the positive observed $O_3MDA8$ trend in areas of China with a $NO_x$ limited or mixed Ozone regimes that cover the majority of China (Jin and Holloway, 2015)."

*168. The Li et al. (2019a) study blames heterogeneous chemistry happening in declining particles for the negative trend. Is this process included in this WRF-Chem configuration and how this influences your results? An attempt to compute similar metrics as in the Li study might be good to intercompare results.*

In our version of WRF-Chem, the heterogenous $HO_2$ uptake process is not included. Li et al. (2019b) conclude that this process is the main driver for the positive $O_3$ trend across China, which has been disputed in Tan et al. (2020), who find that this process was not significant in the NCP region. The main aim of our study was to compare the importance of meteorology and emissions in driving trends of major air pollutants. Therefore, we believe that performing additional model runs to specifically examine the chemistry driving the $O_3$ trend is beyond the scope of this study.

*191-196. Is not clear in this paragraph where you consider the model issues on O3 trends.*

In this paragraph, we acknowledge the underestimation of the positive $O_3$ trend, which is why we do not use the modelled trends to estimate the change in $O_3$ caused mortality. We instead apply the measured trend to the 2015 model fields. We have clarified this in the paper.

*Minor Edits*

*75. I believe ACP policy is to not use links but references, please check.*

We have seen these data sources hyperlinked in other ACP articles, and will verify this is acceptable during the editing process.

*124. Fix issue with the symbol after 0.05*

This has been corrected

*329. Should this be 2019b?*

The references have been checked to ensure they correspond to the correct paper.

**References used in response document**

[revised manuscript text omitted]

---

## Author Response (AR2)

**Response to comments on the revised submission of 'Pollutant emission reductions deliver decreased PM2.5-caused mortality across China during 2015–2017'**

**Response to Editor comments:**

*Concerning the data, I would recommend that you store the data behind your study on an open-accessible repository (with a DOI). The current way of referencing to data via web addresses is not optimal. Please have a look at the data guidelines of ACP: https://www.atmospheric-chemistry-and-physics.net/about/data_policy.html*

We thank the editor for this suggestion and hope that it will be useful to share our data. We have deposited the data in 'Research Data Leeds Repository,' which is registered with https://www.re3data.org/. The repository has the DOI: https://doi.org/10.5518/878. Here we have included the calculated trends from the measurement data and both model runs, at each of the measurement station locations. We also include a copy of the measurement dataset containing data from 2014-05-13 to 2020-06-06, along with the results of our data cleaning process. We think having the measurements dataset in this convenient and accessible form will be very useful to other researchers. We have amended the data availability statement accordingly.

[Figure]

*One further question from my side: In Sect. 2.1, you write "We conducted quality control on the measured data following the methods outlined in Silver et al. (2018), which include excluding data with a high proportion of repeated measurements and periods of low variability." Why do you exclude data with low variability? Shouldn't it be the opposite (e.g. excluding local plumes/spikes in the observational data)?*

The periods of low variability we refer to are periods where almost exactly the same measurements are repeated each day for long periods. This example shows the anomalous data found at some stations which we wish to exclude. The algorithm highlights the area of almost repeated measurements. If the station has more than 60 days of flagged data, we exclude it from the analysis. The algorithm is

described in our previous paper ([https://doi.org/10.1088/1748-9326/aae718](https://doi.org/10.1088/1748-9326/aae718)). We have amended lines 85-87 to better descript this as "We conducted quality control on the measured data following the methods outlined in Silver et al. (2018), which include excluding data with a high proportion of repeated measurements identified as periods of low variability, which represent periods of missing or invalid data."

**Response to referee #2**

*The authors really worked hard on the new version of the manuscript considering comments from all reviewers. I believe the manuscript is in much better shape now and I only have minor comments.*

We thank the reviewer for the positive comments.

*Comments by line:*

*115-128. This paragraph corresponds to model evaluation, not to trend estimation, so consider having it in it's own section (maybe even move it to results)*

We thank the referee for pointing this out. We have moved that paragraph as suggested to the next section, and given it the sub-heading '3.1 Model Evaluation.'

*Section 3.1. It would be informative to add the trends in aerosol composition for the Spartan Beijing site and compare it to a nearby PM2.5 trends, both in observations and model to add to the discussion.*

The SPARTAN data is very useful in evaluating the ability of the model to capture aerosol speciation. However, the SPARTAN data is of limited temporal resolution and consistency. The SPARTAN data is collected every two weeks, but there are large areas of missing data throughout the time series. As a result, trends calculated from the SPARTAN data cannot be confidently compared with those calculated from the CNEMC data or model data. For this reason we would prefer not to add trends from the SPARTAN data to our paper.

*159-161. Another reason could be linked to the representation of aerosol speciation, the fact that NO2 trends are all negative and the model tends to overpredict the nitrate fraction could drive the PM2.5 trends. Same for overprediction of the magnitude of the SO2 trend.*

We agree with the referee that inaccuracies in the model nitrate and sulphate estimations will affect the overall $PM_{2.5}$ trend, among other model inaccuracies. However, due to the lack of speciated aerosol data across China, it is difficult to quantify the degree of inaccuracy in the $PM_{2.5}$ across the model domain. To further highlight this issue, we add the sentence 'However, as the above comparisons with speciated aerosol measurements show, the underlying trends in individual aerosol species may contain inaccuracies that affect the overall $PM_{2.5}$ trend.'

*Fig S4. Add to the caption for what period of time you are comparing*

We have added 'during 2015-2017' to the caption.

*Fig S5. Are statistics build with hourly or daily values?*

According to the Zhou et al. paper, the aerosol components were measure at around 15 minute intervals. We have added the text 'The measurements had a time resolution of ~15 minutes and averaged by season' to the caption.

*Minor Edits*

*69-72. Since these are previous studies you could change the tense to past?*

We have amended these sentences to past tense, thank you for the correction.

[revised manuscript text omitted]